
# Spatiotemporal Changes of Heat Waves and Extreme Temperatures in Main Cities of China from 1955 to 2014

Kuo Li, Gyilbag Amatus

Institute of Environment and Sustainable Development in Agriculture, Chinese Academy of

Agricultural Sciences, Beijing 100081, China

Kuo Li, corresponding author Email: hqlk2000@163.com

**Abstract:** In the past decades, severe heat waves have frequently occurred in many parts of the world. These conspicuous heat waves exerted terrible influences on human health, society, economy, agriculture, ecosystem and so on. Based on observed daily temperatures in China, an integrated index of heat waves and extreme temperature days was established involving the frequency, duration, intensity, and scale of these events across large cities in China. Heat waves and extreme temperature days showed increasing trend in most regions except Northwest China from 1955 to 2014. After late 1980s, the increasing trend was more obvious than the past decades. The cities in the middle and lower reaches of the Yangtse river were threatened by the most serious hot events in the past 60 years, especially Chongqing and Changsha. Due to the subtropical monsoon climate and special terrain, Chongqing would occupy the top of hot cities in a long period. In particular, there was obvious fluctuation for hot years in 31 cities, which were not continuously rising with the global warming; 21 cities mainly located in the eastern and southern regions of China had obvious rising trend; 8 cities had clear declining trend which mainly distributed in the western and northern regions of China; and there were no extreme temperature days in Kunming and Lasa in the past 60 years. The study revealed an obvious differentiation of hot events for 31 cities under climate change; hot threat in most cities is aggravating but declining or remained unchanged in the other cities. The trend is likely to intensify with global warming.

**Keywords:** Heat waves, Extreme temperature days, Hot Year Index, Climate change, China

## 1 Preface

In the past 100 years, global warming has been an apparent physical phenomenon in the



whole world (Stocker et al, 2013). The extreme events (heat waves, flood, drought, typhoon)
frequently break out in many parts of the world, which exert huge effects on normal
functioning of agriculture, society, human health and ecosystem (Alexander et al., 2006;
Diffenbaugh et al., 2016; Coumou and Rahmstorf, 2012). In the past decade, heat waves (HWs)
engulfed many countries worldwide, impacting negatively on the whole population especially
the elderly and children (Horton et al., 2015; Liu et al., 2012; Angélil et al., 2017; Peterson et
al., 2013); for example, in 2003, the European continent experienced an extraordinary HW
which was characterized by excessive long duration, unprecedented extreme temperature
and vast spatial scale. This devastating HW took a heavy toll on human lives (at least 50,000
deaths) (Stott et al, 2004; Robine et al., 2008). In 2013, a similar HW visited most parts of
China with increased intensity and duration resulting in significant economic loss (Sun et al.,

2014).

Concrete definition and exact assessment of HW has become the main obstacles in
developing mitigation and adaptation measures (Hajat et al., 2006; Perkins and Alexander,
2013). A HW is usually defined as an event that exceeds prescribed temperature thresholds
over a few days (Robinson, 2001). Precise definitions are created in many literature which pay
attention to different features of HWs (Bonsal et al., 2001; Klein et al., 2003; Jones et al., 2015).
Climate scientists attach greater importance to how to evaluate the intensity and frequency
of HWs; disaster scientists pay more attention to the vulnerability evaluation and risk
assessment of HWs; sociologists mainly focus on the human health impact of HWs which
attempts to estimate the probable heat-related mortality and morbidity of people; besides,
there are many researchers who focus on the impact of HWs on agriculture, water resources,
forestry, ecosystem and other sectors (Dike et al., 2015; Johnson et al, 2009; Dong et al., 2015;
Buscail et al., 2012). On the whole, there are two research trends for HWs; one is about the


characteristics analysis of HWs; the other is about the impact assessment and consequence
analysis of HWs. The feature analysis of HWs is the basis for impact assessment on different
sectors (Liang et al., 2014; Fouillet et al., 2006). But the realities of HWs in different continents
are distinctive, so the definitions and thresholds of HWs are debatable for researchers.
In Canada and USA, the HW threshold is 40.5℃; when the time is more than 3 hours
accumulated in 2 days in which the temperature is over 40.5℃, a HW could be confirmed; the
other threshold of HW is 46.5℃, over which in any time of a day a HW would be confirmed
(Oswald et al., 2014). In the Netherlands, the HW refers to a period of at least 5 days in which
the extreme maximum temperature (Tmax) in each day exceeds 25℃; in the meantime, the
Tmax exceeds 30℃ in at least 3 days of the above period (Uhe et al., 2016). For World
Meteorological Organization (WMO), the threshold of HW is 32℃, which should be exceeded
in at least 3 days (Klein et al., 2009). In China, a HW usually refers to a period of at least 3 days
when the extreme maximum temperature (Tmax) in each day exceeds 35℃ (Liu et al., 2017;
Chen et al., 2014). In China, the early warnings of HWs are gradually advanced with the
intensity levels of HWs; when Tmax exceeds 35℃, the local meteorological departments
would issue a Yellow Warning; when Tmax exceeds 37℃, the local meteorological
departments would issue an Orange Warning; when Tmax exceeds 40℃, the local
meteorological departments would issue a Red Warning. In a comprehensive view, the
thresholds of HW in different regions are depending on the local climate conditions.
Unlike US and Europe, HWs assessment in China is primarily focused on occurrence
frequencies of individual warm days with extreme temperatures (Huang et al., 2010; Zhang et
al., 2005). The basic features of other equally important aspects for HWs, such as duration and
intensity, are less emphasized (Li et al., 2010). Some recent studies in the US and the Europe


began to separately assess diverse HW types (Gasparrini et al., 2015; Easterling et al., 2016),
in which the temperature variable (Tmax or Tmin) was delimited into different categorizations
but few studies have been able to integrate the different features of HWs for a holistic
assessment. An integrated index is therefore desirable for systematic and quantitative
evaluation of HWs in China, which includes multiple indicators – frequency, duration, intensity
and so on. Moreover, current definition of HWs in China only considered the thresholds of
Tmax, which is not enough for the precise assessment of HWs. For example, it is hard to
evaluate the exact difference between a HW event (exceeding 35℃, 5 days) and the other HW
event (exceeding 40℃, 3 days). For both scientific literatures and operational practices in
China, it just shows the qualitative situation of scorching conditions, which would not easily
give policy-makers and general public a clear picture of HWs for efficient precautions. As such,
a more quantitative and precise evaluation should be done to distinguish different impacts of
HWs, such as, human health, water resource supply burden, forest fires, ecology degeneration,
among others.

This study therefore aims at building an integrated index of HWs and extreme

temperature days. It would compare the observed basic features of HWs and extreme
temperature events in the typical 31 cities of China during 1951-2014 and reveal the change
trends of HWs and extreme temperature events in mainland China under climate change.
Spatial distribution of HWs and extreme temperature days in the past 60 years in different
cities would be estimated and mapped. The integrated index of HWs and extreme
temperature events would provide an efficient tool for risk assessment of hot events under
future climate change scenarios and support for further physical interpretation, attribution
and mechanism of HWs.
**2 Data and Methods**


## 2.1 Data


Data from the National Meteorological Information Centre (NMIC) of the China
Meteorological Administration (CMA), which is the first and most authoritative national
homogenized temperature data set in China, was used. A database from 31 capital cities in all
the provinces of China with historical daily temperature data from 1951 to 2014 was used,
except Taiwan, Hongkong and Macao. At some stations the daily data was missing, especially
in the years prior to 1955. In order to ensure consistency of temperature extremes and
efficiency of the entire study, missing data up to 2% of the data points at each station in more
than 50 years was rejected. The data of 31 stations over the period from 1955 to 2014 were
ultimately selected for analysis.

## 2.2 Study area


According to the temperature and precipitation data, combined with the administrative
boundaries of provinces, the whole China could be divided into 8 climate regions, including
Northeast China (NE), North China (NC), East China (EC), South China (SC), Southwest China
(SW), Northwest China (NW), Central China (CC) and Qinghai-Tibet Plateau (QT). Locations for
the 31 cities and the climate zones in the study are presented in Fig.1. The total population of
31 capital cities currently stood at 278 million representing 20% of the total population of
China and contributing 33.5% of the country's GDP. These 31 capital cities were therefore
chosen to reveal the trends of extreme temperature in China, which may influence policy
directions in reducing extreme temperature disasters, protecting human health and
enhancing crop production.

## 2.2 Method


In this study, an integrated index is established for systematical and quantitative
evaluation of HWs and extreme temperature events in China, which includes the frequency,



duration and intensity of HWs and extreme temperature days. At first, we made clear two
definitions: extreme temperature days and heat wave (HW). As stated earlier, when Tmax
exceeds 35℃, it could be called a day with extreme temperature in China; when Tmax exceeds
35℃ in more than 2 consecutive days, it could be defined a heat wave (HW) event. The
extreme temperature days are the base of a HW. In one year, there may be several HW events
and discontinuous days with extreme temperature, which jointly decide the hot level of one
region. So the integrated index would contain two aspects in this study, HWs and discrete days
with extreme temperature.

According to the statistical data, the hot days with extreme temperature usually

concentrate on June, July and August in China, which account for above 90% of all the hot
days from 1955 to 2014 in 31 capital cities. In May and September, the hot days account for
9% and in the other months it accounts for no more than 1% (Fig.2). It is obvious that HW
events mostly break out in June, July and August, which are the hottest months of the whole
year in 31 capital cities. So we take the three months as the basic period for intensity
assessment of HWs and extreme temperature days. There are totally 92 days in June, July and
August. If one HW event lasts for 92 days in a year, it would be regarded as the most serious
heat event.
**2.2.1 Heat wave index**

For HW events, the frequency, duration and intensity should be considered. Firstly, if the

HWs last for more days, the intensity of HWs would be bigger. Secondly, according to the
definition of HW,

3 days are the shortest duration for HWs, in which daily Tmax exceeds 35℃. So the period

of 3 days is made as one essential unit for evaluating the intensity of HWs. Thirdly, as
mentioned above, when daily Tmax exceeds 37℃ or 40℃, especially the continuous days





above 37℃ or 40℃ are increasing, the intensity of HWs would go up rapidly. So in the study,
Heat Wave Index (HWI) is established as the following formula.
$$\mathbf{HWI} = \left(\frac{\mathbf{CD_{35}}}{\mathbf{92}} \times \frac{\mathbf{CD_{35}}}{\mathbf{3}} + \mathbf{1}\right) * \left(\frac{\mathbf{AD_{37}}}{\mathbf{92}} + \frac{\mathbf{CD_{37}}}{\mathbf{3}} + \mathbf{1}\right) * \left(\frac{\mathbf{AD_{40}}}{\mathbf{92}} + \frac{\mathbf{CD_{40}}}{\mathbf{3}} + \mathbf{1}\right) \qquad (1)$$

HWI represents the integrated intensity of HW events: CD35 represents the continuous
days in which daily Tmax exceeds 35℃; AD37 represents the all days in which daily Tmax
exceeds 37℃ among CD35; CD37 represents the continuous days in which daily Tmax exceeds
37℃ among CD35; AD40 represents the all days in which daily Tmax exceeds 40℃ among CD35;
CD40 represents the continuous days in which daily Tmax exceeds 40℃ among $CD_{35}$.
For one year, there may be several HW events. The total intensity of Annual HWI (AHWI)
should contain all HW events of the year. Based on HWI, AHWI is calculated as following.
$$\mathbf{AHWI} = \sum_{\mathbf{i=1}}^{\mathbf{n}} \mathbf{HWI_i} \qquad (2)$$

AHWI represents the total annual intensity of HW events; n represents the total
frequency of HW events in one year; i represents the sequence of HW events occurred in one
year.
**2.2.2 Hot year index**
As mentioned above, within one year, there are not only HW events, but also
discontinuous days with extreme temperature. If the hot levels are compared between
different cities in different years, the two aspects should be considered synthetically. The
discontinuous days with extreme temperature above 35℃, 37℃ or 40℃ are not as serious
as HW events in some cities. In other cities there may be few HW events in some years, in
which the hot levels are mainly decided by the discontinuous days with extreme temperature.
So based on AHWI established above, an integrated index for hot years is constructed,
considering the discontinuous days with extreme temperature in one year. The formula is as
follows:
$$HYI = AHWI + \frac{D_{35} - \sum CD_{35}}{92} \times \frac{D_{35} - \sum CD_{35}}{3} + \frac{D_{37} - \sum AD_{37}}{3} + \frac{D_{40} - \sum AD_{40}}{3} \quad \textbf{(3)}$$

HYI represents the integrated intensity of hot years in different cities. $D_{35}$ represents
the days of one year in which daily Tmax exceeds 35℃; $\sum CD_{35}$ represents the continuous
days in which daily Tmax exceeds 35℃ in one year; $D_{37}$ represents the days in one year in
which daily Tmax exceeds 37℃; $\sum AD_{37}$ represents the all days in which daily Tmax exceeds
37℃ among $CD_{35}$ in one year; $D_{40}$ represents the days in one year in which daily Tmax exceeds
40℃; $\sum AD_{40}$ represents the all days in which daily Tmax exceeds 40℃ among $CD_{35}$ in one
year.
**3 Results**
**3.1 Trends of Extreme Temperature days**
According to the historical statistics, Chongqing has been threatened by the most serious
disasters of extreme temperature in whole China, in which annual $D_{35}$ exceeds 33 days in the
past 60 years. Meanwhile, there is no extreme temperature day from 1955 to 2014 in Kunming
and Lasa, which are the most comfortable places of the 31 capital cities in summer. There are
7 cities in which annual $D_{35}$ is between 20-30 days (Fig.3), including Changsha, Fuzhou,
Nanchang, Hangzhou, Haikou, Xi'an and Wuhan. With regards to climate zones, Central China
had been threatened by the most frequent extreme temperature disasters in the past 60 years;
annual D35 in East China and South China was between 10-20 days; North China and
Southwest China was between 1-12 days; Northwest China was about 8 days; and Northeast
China and Qinghai-Tibet Plateau was less than 3 days.
Though the global climate has been continuously warming in the past 60 years, the trend
of $D_{35}$ in 31 main cities of China is not increasing constantly. There are 3 main stages for the





variation of $D_{35}$ in China (Fig.4). From 1955 to early 1970s, the value of $D_{35}$ in 31 cities of China
averagely amounts to 372 days per year, signifying the high level of hot years in this stage;
from early 1970s to late 1980s, the value of D35 in 31 cities of China averagely amounts to
280 days per year, which means that, these cities encountered a relatively cool years in this
stage; from early 1990s to 2014, the value of D35 in 31 cities of China averagely amounts to
425 days per year, which is higher than the past 40 years. It means that the whole China is
threatened by more and more serious extreme temperature events in the recent 20 years.
However, there are obvious variation in the characteristics of D35 in different climate zones
of China. The values of D35 in South China, East China and Northeast China are obviously going
up from 1955 to 2014; the values of D35 in Central China, Southwest China and North China
are slightly rising; however, the trend in the values of D35 in Northwest China have slightly
declined in the past 60 years.
**3.2 Trends of Heat Waves**

Following the HWs definition in China, an average of 1.54 HW events occurred annually

in each city from 1955 to 2014, which last for an average of 5.4 days for each HW event. It is
obvious that, as the value of D35 gets bigger in each city, the amount and frequency of HWs
also grow bigger (Fig.5). There is a positive correlation between D35 and HWs. Through the
analysis of HWs in the 31 typical cities, Chongqing was the most threatened as HW rose up to
25.1 days annually; Changsha experienced the most frequent HWs in the past 60 years, almost
3.9 times per year; the intensities and frequencies of HWs in Nanchang, Fuzhou, Hangzhou,
Haikou and Xi'an are smaller than Chongqiang and Changsha, but much bigger than other
cities; there was no HW in Kunming, Lasa and Changchun but there were few HWs in Haerbin,
Shenyang, Guiyang and Xining. For the other cities, the threat from HWs was in the middle
level.



According to the statistics, the distribution of amounts and frequencies of HWs per year
in the 31 cities were similar to the distribution of D35 (Fig.6). Comparing the different climate
zones, cities in Central China had been threatened by the most serious HWs in the past 60
years, in which the frequency and amount of HWs per year were the highest; in cities of East
China HWs have also been very serious; in cities of South China and Southwest China the
threat of HWs have been lower than the Central China and East China; in cities if North China
and Northwest China there were less annual HWs; in cities of Northeast China and Qinghai-
Tibet Plateau, there had been almost no obvious threat of HWs in the past 60 years.
**3.3 Heat Wave Index**
In order to do comparative analysis on the HWs occurrence in the different cities for the
past 60 years, a Heat Wave Index (HWI) was established as mentioned above. The duration
and intensity are the key factors of HWs that define the severity of hot events. So HWI is
designed to refer to the number of days one HW event lasts and the maximum temperature
one HW event reaches (Tab.1). HWI provides us a quantitative tool to distinguish the different
HWs in 31 typical cities of China. According to the climate conditions and national standards
of extreme temperature in China, HWs could be classified into 5 levels of hazard by the values
of HWI. When the value of HWI is 1.0, it indicates that there is no hot day in which $T_{max}$ exceeds
35℃. When the value of HWI is between 1.0 and 1.5, it indicates slight HW hazards in which
the duration and intensity of HWs are minimal. When the value of HWI is between 1.5 and
3.0, it means HW hazards are not serious as there are few days of Tmax exceeding 37℃. When
the value of HWI is between 3.0 and 6.0, it indicates that the HW hazards are serious and the
days of $T_{max}$ exceeding 37℃ or 40℃ become frequent. When the value of HWI is above 6, it
indicates that the HW hazards are very serious and the days of $T_{max}$ exceeding 37℃ or 40℃
may last through the whole period of HWs.



According to the classification of HWI, the frequencies of HW hazards with different levels
in the past 60 years in 31 typical cities of China are analyzed (Fig.7). In all, cities with low HW
hazards were the majority accounting for 62.3% of all HWs; the moderate HW hazards
accounted for 26.4%; the high HW hazards represented 7.7%; and the extreme high HW
hazards accounted for 3.6%. For all the 31 cities, most of the HW hazards are not serious; only
1/10 of the HW hazards are of the greatest threats. No HW hazards occurred in Changchun,
Shenyang, Guiyang, Kunming and Lasa from 1955 to 2014; no high or extreme high HW
hazards occurred in Haerbin, Huhehaote, Xining, Yinchuan, Taiyuan and Chengdu; no extreme
high HW hazards occurred in Beijing, Tianjin, Wulumuqi, Lanzhou, Guangzhou and Nanning;
in the remaining 14 cities, there were all four levels of HW hazards occurred in the past 60
years. However, most HW events of high (0.57 per year) and extreme high (0.47 per year)
levels occurred in Chongqing than the other cities; most HW events of moderate levels
occurred in Xi'an, reaching 1.35 per year; and most HW events of low level occurred in Haikou,
reaching 2.38 per year.
Based on the calculation of HWI, the sum of HWIs from 1955 to 2014 in each city is shown
in Fig.8. It is obvious that Chongqing has been threatened by the most serious HW hazards in
the past 60 years, in which the frequency, duration and intensity of HWs are the biggest of all
the 31 cities. The sum value of HWIs in Chongqing is far bigger than other cities; the annual
sum value of HWIs in Chongqing reached 13.7. Changsha had been the second hard hit city
with most serious HW hazards, in which the annual sum value of HWIs reached 9.5. There
were 6 cities that have been threatened by severer HW hazards, include: Hangzhou, Fuzhou,
Nanchang, Xi'an, Wuhan and Haikou; the annual sum value of HWIs in each city is between 4
and 9. There were 7 cities threatened by moderate severe HW hazards; these cities include:
Hefei, Zhengzhou, Nanjing, Jinan, Shijiazhuang, Nanning, and Shanghai and the annual sum





value of HWIs in each city is between 2 and 4. The remaining 11 cities encountered lighter
serious HW hazards in which the annual sum value of HWIs is between 0 and 2. As mentioned
above, there were no HW hazards in 5 cities.
**3.4 Hot year Index**
Based on Heat Wave Indexes, Hot Year Indexes in the 31 cities were calculated and
analyzed, including HW events and discontinuous days with extreme temperature (Tab.2). The
analysis revealed the heat levels of the cities in different years. In the study, the quantity of
Hot Year Indexes for all cities added up to 1860 from 1955 to 2014.
The No-hot year represented 29.1% of the gross; Light hot year, 28.8%; Mild hot year,
20.3%; Moderate hot year, 13.7%; Serious hot year, 7.9%; and the Extreme hot year
represented 0.3%. Chongqing has been threatened by the most severe heat, in which Serious
hot year and Extreme hot year accounted for 50% of the 60 years; in Changsha, Nanchang,
Hangzhou and Fuzhou, Serious hot year and Extreme hot year accounted for 25%. However,
there was only slight heat threat or no heat threat in the past 60 years in most cities of
Northeast China, Northwest China, Southwest China and Qinghai-Tibet Plateau, in which No-
hot year and Light hot year accounted for more than 90%. For the remaining 14 cities, Mild
hot year and Moderate hot year accounted for the most of 60 years. It is obvious that the west
and north regions of China are much cooler than the east and south parts of China; the hottest
regions are located in Central China and East China.
On the point of time series, there are 3 kinds of variation trends of HYI for all the 31 cities:
uptrend, downtrend and no change. In 21 cities, the value of HYI had obvious rising trend; the
remaining 8 cities had clear declining trend in the value of HYI. There were no extreme
temperature days in Kunming and Lasa in the past 60 years, so there was no change of HYI in
the two cities. There are two rising pathways for the 21 cities; one is rising directly; the other



is firstly declining and then rising. In a comprehensive view, there are 3 stages for all the cities
in the past 60 years. In the first stage from 1955 to the early years of 1970s, HYIs in most of
cities were in a high level; the moderate hot years and serious hot years were frequent, which
accounted for 27.0% of the first stage. In the second stage from the middle of 1970s to the
end of 1980s, HYIs in most of the cities were in a low level; the moderate hot years and serious
hot years were rare, which accounted for 11.7% of the second stage. In the third stage from
the early years of 1990s to 2014, HYIs in most of cities were also in a high level; the number
of moderate hot years and serious hot years accounted for 26.8%; but the severities of hot
years in this stage are more serious than the first stage in most cities. In general, there was
obvious fluctuation for hot years in the past 60 years in the 31 cities, which are not
continuously rising with the global warming. There was obvious increasing trend for whole
China, either the intensity or the frequency of HWs and extreme temperature days.
From figure 9, clear variations of HWI events existed in most cities across the main land
of China. For example, in Northwest China, HYIs in Lanzhou and Yinchuan were so small that
no serious hot events occurred in the past 60 years, but in Wulumuqi and Xi'an, HYIs were
much pronounced as annual average value of HYIs from 1955 to 2014 in Xi'an reached 6.96.
In North China, the annual average values of HYIs in Beijing, Tianjin and Taiyuan were between
1.2 and 2.4, in which light hot years represented 63% of the whole; but in Shijiazhuang, Jinan
and Zhengzhou, the annual average values of HYIs were between 3.9 and 5.1 and mild hot
years represented 43% of the whole. In Southwest China, there were few hot waves in
Chengdu, Guiyang and Kunming making these cities as cool as Northeast China; however, in
Chongqing, the annual average value of HYIs rose up to 15.0. This city had been threatened
by the most severe hot events, as serious hot years represented 34% and the HYIs ranked first
of the 31 cities in 27 years of the past 60 years. From a broader view, 3 types of regions were




identified: Northeast China and Qinghai-Tibet    Plateau composed of one type of the regions:
HYIs of these cities were small and the annual average value was 1.02 in which No-hot years
accounted for more than 60%, representing the coolest region in China; Central China, East
China and South China also formed one type of regions: HYIs of most of these cities were
higher than the other regions and the annual average value of HYIs rose up to 5.61, in which
moderate hot years and serious hot years accounted for 40%; in Northwest China, Southwest
China and North China which formed the last type of the regions, HYIs of most these cities
were in the middle and the annual average value of HYIs was 3.45, in which light hot years and
mild hot years accounted for 54%.
In brief, there is an apparent feature that most of the cities that were threatened by
serious hot events in the past 60 years gather in the middle and lower reaches of the Yangtse
river; there were few hot events in NE, NW, SW and QT, except Chongqing, Xi'an and Wulumuqi;
the threatened by hot events in SC is not striking, though the annual mean temperatures of 3
typical cities in this region is the highest of all 31 cities.
**4 Discussion**
With global warming, there have been a lot of researches focusing on HWs. Most of these
studies paid more attention on a single factor of HW, especially on occurrence frequency. The
other key indicators, such as duration, intensity, extent and timing, were usually neglected.
There are few studies combining HWs with extreme temperature days to evaluate the annual
hot events and compare the inter-annual changes of torridity degrees.
From our analysis, we established a statistical model involving the frequency, duration,
intensity, and length of the HWs and extreme temperature days across large cities in China.
By analyzing HWs and extreme temperature days in large cities of China, we are capturing the
changes and spatial distribution in HWs and the extreme temperature events caused from


climate fluctuation and climate change, as well as local changes from the urban environment.
The results presented in this study are consistent with previous findings on changes in
extreme temperature days and HWs in recent decades across China due to global-scale drivers
(Chen et al., 2017; Fang et al., 2016; You et al., 2013; Qi et al., 2012). HW is the basic element
for evaluation of hot events which is taken into account in most of the researches across the
whole world (Spinoni et al., 2015; Oswald et al., 2014; Santamouris et al., 2015; Gershunov et
al., 2009). However, the discontinuous extreme temperature days are usually ignored which
play an important role on evaluation of annual hot events. The common influences caused by
HWs and extreme temperature days exhibit the overall scene of hot events in different cities.
The increase in the number of HWs and extreme temperature days in China, are consistent
with all other global or regional studies that show that the occurrence of warm days increased
(Rusticucci et al., 2012; Nemec et al., 2013; Pingale et al., 2014). The abrupt changes in the
trends of HWs and hot years mainly occurred in the 1970s and 1980s; there was a period from
early 1970s to late 1980s, in which the number of HWs and extreme temperature days were
relatively lower than the other years; the changes are in accordance with the former findings
put forward by other researchers (Zhou and Ren, 2011; Xu et al., 2013).
The cities distributed in the middle and lower reaches of Yangtze River had been
threatened by the most serious HWs and hot years in the past 60 years, especially Chongqing
and Changsha. The long-term anticyclones and the special topography are most responsible
for this trend of change; Chongqing is a located in a valley surrounded by mountains and
Changsha is located in the valley of Xiangjiang river, which are both affected by subtropical
monsoon climate. At the mean time, the location, scope and intensity of HWs and extreme
temperature events in southern China are closely influenced by the western Pacifica
subtropical high and the East Asia jet stream (Wang et al., 2015). In North China, the threat by



HWs and hot years in the past 60 years is relatively mild, except Xi'an and Zhengzhou. The
main cause is due to the anticyclone circling over the Lake Baikal (Ding et al., 2010). For most
cities in western and northern China, the high latitudes and high altitudes remarkably restrict
the occurrence of HWs and extreme temperature events, in which the threat is slight and
there is no obvious increase in the past 60 years (Zhou and Ren, 2011). It is therefore
worthwhile to explore how the atmospheric circulation patterns change in future which would
reveal the spatiotemporal trends of HWs and extreme temperature events in China. On the
other hand, the elaborate depiction and accurate evaluation of HWs and extreme
temperature events in more cities of China would be meaningful for planning of disaster
prevention and mitigation.
**5 Conclusions**
This study established an integrated index which contained the duration, intensity, extent
and timing of HWs and extreme temperature days. It showed the whole picture of hot threat
in 31 main cities from 1955 to 2014.
(1) Both HWs and extreme temperature days showed increasing trend from 1955 to 2014 in
NC, CC, NE, SW, EC and SC; there was a slight decreasing trend in cities distributed in NW.
For whole China, HWs and extreme temperature days exhibited an obvious upward trend
in the past 60 years with a rapid increase after late 1980s.
(2) The hottest cities were located in CC and EC over the past 60 years; the cities in SC and NC
were faced with middle level of threat; there were low threat of heat events in most of
the cities from NE, NW and SW, except Chongqing and Xi'an. More especially, Chongqing
had been threatened by the most serious HW hazards, much heavier than the other cities.
(3) There was obvious fluctuation for hot years in 31 cities over the past 60 years, which were
not continuously rising with the global warming; 21 cities mainly located in the eastern




and southern regions of China had obvious rising trend; 8 cities had clear declining trend

which mainly distributed in the western and northern regions of China; however, there

were no extreme temperature days in Kunming and Lasa in the past 60 years. More

specially, there were 3 stages for all 31 cities and the abrupt changes occurred separately

in early 1970s and late 1980s.

**Acknowledgments:** This research is supported by the National Key R&D Program of China (Project No. 2017YFD0300301) and National Natural Science Foundation of China (Project No.41871026). Special thanks are due to the National Meteorological Information Centre (NMIC) of the China Meteorological Administration (CMA) for providing the national homogenized temperature data set in China.

**Data availability:** The historical weather data (1955-2014) that support the analysis in this study is from the National Meteorological Information Centre (NMIC) of the China Meteorological Administration (CMA), which is publicly available online at http://data.cma.cn/.

**Author contribution:** The first and corresponding author (Kuo Li) is in charge of the data analysis, model construction and writing. The second author (Gyilbag Amatus) is responsible for data collection, mapping and polishing.

**Competing interests:** we declare no competing interests in this article.

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

**Tab.1 The classification of HW hazards by the values of HWI**

| Heat Wave Index | Level of hazard | Description |
|---|---|---|
| HWI =1.0 | No hazard | There is no HW event occurred. |
| 1.0＜HWI≤1.5 | Low hazard | The HW event must last at least 3 days and less than 12 days, in which there is no continuous days above 37℃ or 40℃. |
| 1.5＜HWI≤3.0 | Moderate hazard | The HW event must last at least 3 days and less than 24 days, in which there is at most 5 continuous days above 37℃. |
| 3.0＜HWI≤6.0 | High hazard | The HW event must last at least 3 days and less than 38 days, in which there is at most 10 continuous days above 37℃. |
| 6.0＜HWI | Extreme high hazard | The HW event must last at least 5 days, in which there is at least 5 continuous days above 37℃. |


**Tab.2 The classification of Hot Years by the values of HYI**

| Hot Year Index | Level | Grades | Description |
|---|---|---|---|
| HYI =1 | No-hot year | 0 | There are neither HWs nor hot temperature days (>35℃) occurred in one year. |
| 1＜HYI≤2 | Light hot year | 1 | There is one HW or a few hot days occurred in one year, which are small and slight. |



| $2<HYI\leq5$ | Mild hot year | 2 | There are a few HWs or hot days occurred in one year, which are usually small. |
| $5<HYI\leq10$ | Moderate hot year | 3 | There are several HWs or some hot days occurred in one year. |
| $10<HYI\leq50$ | Serious hot year | 4 | There are some HWs in high level or many hot days occurred in one year. |
| $50<HWI$ | Extreme hot year | 5 | There are some extreme HWs or a lot of hot days occurred in one year. |



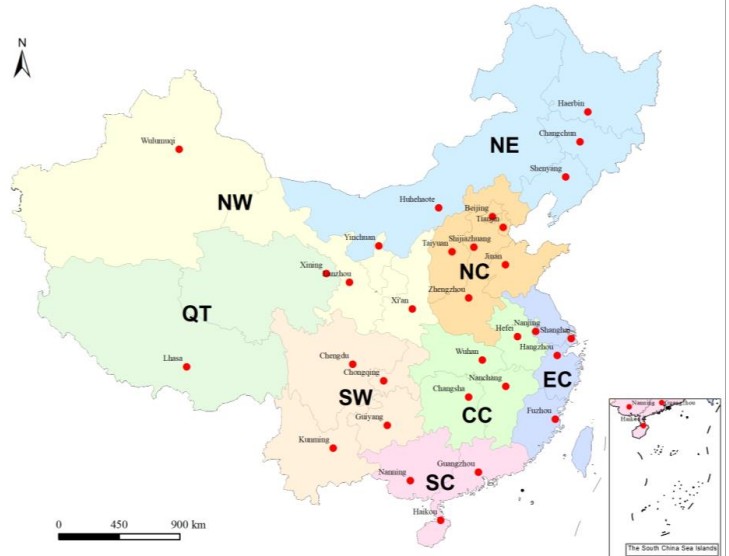


**Fig.1 Distribution of the weather stations in 31 cities and climate zones in Mainland of China (The climate zones includes:**
**NE, NW, NC, CC, EC, SC, SW, QT)**



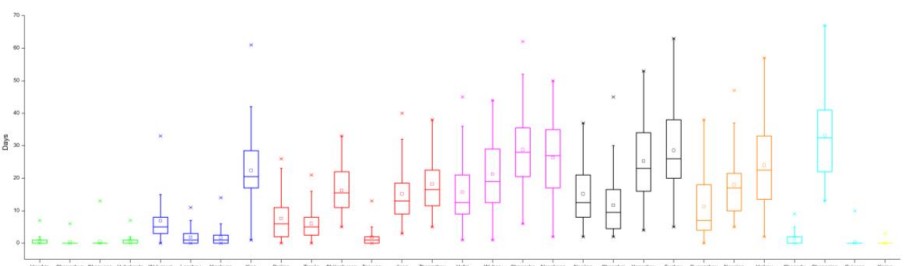

**Fig.2 The proportion distributions of hot days in 12 months from 1955 to 2014 in 31 capital cities in China**

**Fig. 3 Distribution of D$_{35}$ in 31 cities from 1955 to 2014 (Green color: NE; Blue color: NW; Red color: NC; Purple color: CC; Black color: EC; Orange color: SC; Cyan color: SW; Yellow color: QT)**

**Fig. 4 Time series of D$_{35}$ in different climate zones of China from 1955 to 2014**

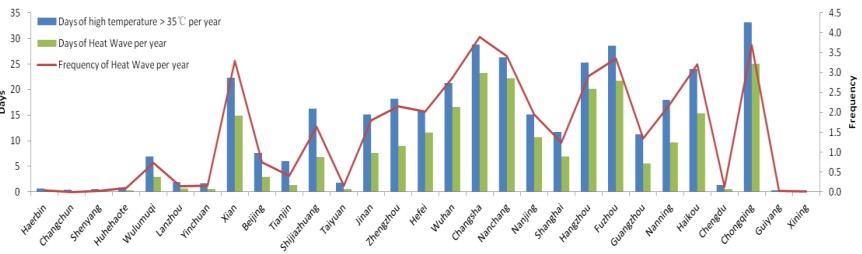

**Fig. 5 Comparison between D₃₅ and HWs per year in 31 cities of China from 1955 to 2014**

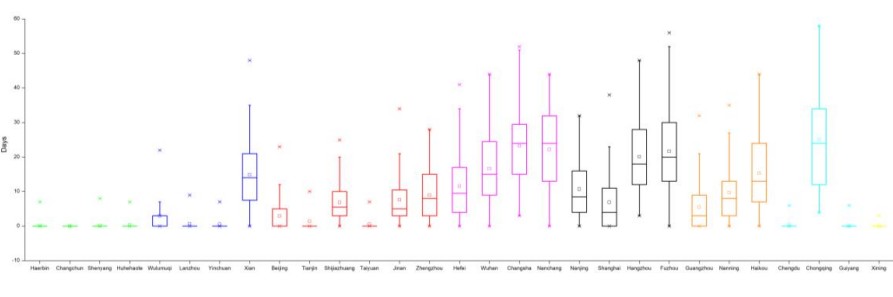



**Fig. 6 Distribution of amounts and frequencies of HWs in 31 cities from 1955 to 2014 (left graph: amounts of HWs; right**
**graph: Frequency of HWs. Green color: NE; Blue color: NW; Red color: NC; Purple color: CC; Black color: EC; Orange color:**
**SC; Cyan color: SW; Yellow color: QT)**

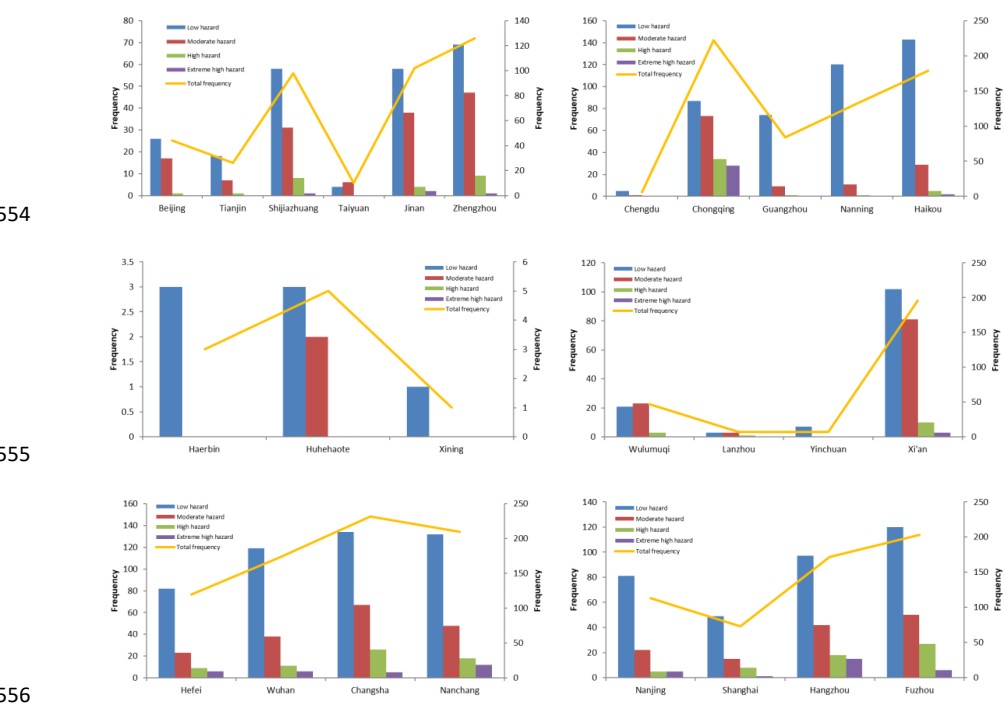





**Fig. 7 Frequency of Low, Moderate, High and Extreme high HW hazards in 31 cities from 1955 to 2014 (Top left: NC; Top**
**right: SW & SC. Middle left: NE; Middle right: NW & QT; Bottom left: CC; Bottom right: EC)**


**Fig. 8 The sum values of HWIs in 31 cities from 1955 to 2014**



**Fig. 9 Classification of Annual Average of HYIs from 1955 to 2014 in 31 cities in Mainland of China**
**(The climate zones includes: NE, NW, NC, CC, EC, SC, SW, QT; The upper blue line: the Yellow River; The below blue line:**
**the Yangtse River)**