# Peer review of "Spatiotemporal Changes of Heat Waves and Extreme Temperatures in"

_Natural Hazards and Earth System Sciences, 2019_

## Referee Comment (RC1) · Anonymous Referee #1 · 6 Feb 2020

The manuscript is interesting, well written, well structured and easily understood by the reader. The authors introduce Heat Wave and Hot Year Indices in order to define and study heat waves in China. The statistical methodology is relatively simple, but useful and leads to interesting conclusions. However, I think that the authors could try to link the trends in heat waves to the corresponding variations/trends in the large-scale temperature and circulation characteristics over the region. For example, they could use grid point temperature, sea-level pressure and/or geopotential height data for the lower troposphere obtained from any of the known worlwide data sets (e.g NCEP/NCAR) in order to further support the justification analysis made in the last paragraph of the discussion section. This would be useful, taking into account that meteorological data

obtained from surface meteorological stations located in (or near) cities are possibly affected by urbanization. I do not suggest an extensive analysis about this issue, taking into account that this case the manuscript would become huge, but 1-2 figures and a paragraph refering to such comparison could be useful.

Minor comments:

1. Page 5, Data and Methods, lines 112-115: Please explain more analytically the procedure used for the division of China into the 8 climate regions, or alternatively provide appropriate references.

2. Pages 6-8, Data and Methods, lines 143-180: Please provide a simple sensitivity analysis of the defined indices. For example, what are the ranges of the indices' values and how are these values affected by specific changes of CD, AD etc. The authors give some answers about this issue later in the results, but in my opinion the methodology section is the proper section to analyze this.

3. Figs 3 and 6: Please explain either in the caption or in the text what are the lines, boxes, points etc. in the diagram.

4. Lines 111 and 122: The number of the titles are the same (2.2). Please correct.

---

## Referee Comment (RC2) · Anonymous Referee #2 · 20 Mar 2020

The manuscript proposes new indices for the study of heat waves and extreme temperature especially for cities in China. I find the manuscript to be interesting and the methodology is novel. Unfortunately, the proposed methodology is not well descripted, and more details should be added. I find the overall information presented in this paper below the standards of the Natural Hazards and Earth System Sciences and I believe that the paper requires entire modifications and needs to go through the review process again. Indeed, there are some aspects that are weak. The main problem is the proposed methodology for the new indices. More specific: • The physical explanation of the index HWI (line 151, page 7) should be added. • I believe that there is a mistake in the equation of the first index, HWI, (line 151, page

7). Is the multiplication sign correct of the first parameter for the CD35? I can not understand why it is multiplication and not sum. The rate CD35/92 should be change to AD35/92. • Moreover, I believe that an example should be added. I was tried to create an example for better understanding. Lets say that in a year there are 35 days with temperature greater than 35oC, from these days, there are 15 consecutive days with temp>35oC. Moreover from the initial 35 days, 15 days have temperatures greater than 37 oC (with 10 consecutive days greater than 37 oC) and 5 days have temperatures greater than 40 oC (with 10 consecutive days greater than 40 oC). Based on these data: HWI=(35/92 x 15/3+1)x(15/92+10/3+1)x(5/92+3/3+1) = HWI=(0.38x5+1)x(0.163+3.333+1)x(0.054+1+1)=2.9x4.496x2.054=26.78 In case there is a mistake in the equation, HWI=6.38x4.496x2.054=58.92

Based on the classification of Table 1, it is obvious that there is a mismatch for the range of the index. Please provide the appropriate modifications and explanations. • In the case of AHWI, there is a misunderstanding. It is not clear, how it is possible to be several HWI in a year. HWI use for its calculation the days with temperature greater than 35 (37/40) in the three months (June, July, August 92 days). Based on it, it is not possible to have more than one value per year. Please give some explanations. • Based on the above comment, HYI (line 173, page 8) can be not defined with the proposed way. Below the Authors can find some minor comments and suggestions in case of resubmission. • Initially, I will suggest the description of the classification of the indices (table 1 and table 2) to be removed into methodology. • The analysis of figure 8 is not consistent with the figure 8. The scale of the diagram in figure 8 range from 0 to 900, the station Chongqing presents HWI equal to 800 while in the manuscript it is said "...sum value of HWIs in Chongqing reached 13.7..." (line 261, page 11). Similarly, the result about Changsha. Please made the appropriate modifications. • The section 3.1 can be changed to "variance of extreme temperature days" since in this paragraph it is analysed the trend of the extreme temperature days but the variance. • The quality of all figures is poor. The labels are too small, and it can not be read. • The authors should add more information about the secondary axis in figure 4 and

7. • The authors claim that the analysis is for 31 main stations in China, in figure 3, 6 and 5 are presented the results of 29 stations, while in figure 7 and 8 are presented the result of 26 stations. Similarly, in map of figure 9 is presented 29 station. Please provide the appropriate modifications.

---

## Author Comment (AC1) · 26 Apr 2020

Dear reviewer,

Thanks very much for the review. The suggestions are very important for our manuscript. We have tried our best to revise the manuscript according your advices and explain the questions as much as we can. The details are listed as following which are also listed in the attached .pdf supplement file.

Best wishes,

Authors

(1) comments from Referees

The manuscript is interesting, well written, well structured and easily understood by the reader. The authors introduce Heat Wave and Hot Year Indices in order to define and study heat waves in China. The statistical methodology is relatively simple, but useful and leads to interesting conclusions. However, I think that the authors could try to link the trends in heat waves to the corresponding variations/trends in the large-scale temperature and circulation characteristics over the region. For example, they could use grid point temperature, sea-level pressure and/or geopotential height data for the lower troposphere obtained from any of the known worlwide data sets (e.g NCEP/NCAR) in order to further support the justification analysis made in the last paragraph of the discussion section. This would be useful, taking into account that meteorological data obtained from surface meteorological stations located in (or near) cities are possibly affected by urbanization. I do not suggest an extensive analysis about this issue, taking into account that this case the manuscript would become huge, but 1-2 figures and a paragraph refering to such comparison could be useful.

Minor comments:

1. Page 5, Data and Methods, lines 112-115: Please explain more analytically the procedure used for the division of China into the 8 climate regions, or alternatively provide appropriate references. 2. Pages 6-8, Data and Methods, lines 143-180: Please provide a simple sensitivity analysis of the defined indices. For example, what are the ranges of the indices' values and how are these values affected by specific changes of CD, AD etc. The authors give some answers about this issue later in the results, but in my opinion the methodology section is the proper section to analyze this. 3. Figs 3 and 6: Please explain either in the caption or in the text what are the lines, boxes, points etc. in the diagram. 4. Lines 111 and 122: The number of the titles are the same (2.2). Please correct.

(2) Author's response:

1. The related reference about the division of China into the 8 climate regions are added in the part of Data and Methods, Pages 5, lines 113-114.

2. We have added a simple sensitivity analysis of the defined indices in the part of Data and Methods, Pages 7-8, lines 156-163 and lines 190-197.

3. The explains of the lines, boxes and points in the caption and the text of Fig 3 and 6 are added in Pages 23-24, lines 566-571 and lines 583-589.

4. The numbers of the titles are corrected in Pages 5-7, line 122, line 142 and line 173.

(3) Author's changes in manuscript

According to the reviewer's suggestion, we try to link the trends in heat waves to the corresponding variations/trends in the large-scale temperature and circulation characteristics over the region. Taking into account that this case the manuscript would become huge, we do not carry out a detailed analysis on the relationship of heat waves and the large-scale circulation; there are many literatures which have done extensive analysis about this issue and revealed the driving factors of heat waves in different regions of China. So we quoted the published results in the part of Discussion, Page 16-17, lines 378-387 to clarify the control factors of spatial distribution of heat waves.

For advice 1, we have cited the reference from Yang QY et al, which introduced the method, principle and process on the division of China into the 8 climate regions. The reference is as follow:

Yang, Q.Y., Wu, S.H., Zheng, D.,: A retrospect and prospect of researches on regional physio-geographical system (RPGS), Geo. Res., 21(4), 407-417, http://doi:10.1080/12265080208422884, 2002.

For advice 2, we have added the ranges of the indices' values and how are these values affected by specific changes of CD, AD etc. The content is as follow:

For HWI, there are two extreme situations. If there are no heat waves in one year, the

value of HWI would be 1. If there are 92 continuous days of a year in which Tmax exceeds 40°C, the value of HWI would reach the biggest, 33792; for the real world, the second extreme situation would rarely occur except extreme catastrophe shocking. According to the statistics from 1955 to 2014 in China, the most serious heat wave event occurred in Changsha city in 2013 for which the value of HWI is no more than 140. The value of HWI is mostly determined by the number of continuous days in which Tmax exceeds 37°C, even 40°C. If the extreme hot days continue longer, HWI would be more serious.

For HYI, there are also two extreme situations. If there are no heat waves or hot days in one year, the value of HYI would be 1. The value of HYI is largely determined by the value of AHWI, which would reach 33792 at most; in other word, the intensity and frequency of heat wave events in one year is bigger, the hot year index would be more severe. There is insignificant impact on HYI for discontinuous days in which daily Tmax exceeds 35°C, comparing with heat wave events. According to the statistics, the hottest year is also in Changsha city in 2013, which contained the most serious heat wave event from 1955 to 2014 in China.

For advice 3, we have added explanation of the boxes, lines and points in the titles of Fig.3 and Fig.6. The content is as follow:

Fig. 3 Distribution of D35 in 29 cities from 1955 to 2014 (Green color: NE; Blue color: NW; Red color: NC; Purple color: CC; Black color: EC; Orange color: SC; Cyan color: SW; Yellow color: QT); Boxes indicate the interquartile spread (25th and 75th quantiles) with the horizontal line indicating the ensemble median and the whiskers showing the extreme range of D35 in 29 cities Notes: There are no high temperature weather in which daily Tmax exceeds 35°Cin Kunming and Lasa cities in the past 60 years. Therefor there are 29 cities shown in this figure.

Fig. 6 Distribution of amounts and frequencies of HWs in 29 cities from 1955 to 2014 (upper graph: amounts of HWs; lower graph: Frequency of HWs. Green color: NE;

Blue color: NW; Red color: NC; Purple color: CC; Black color: EC; Orange color: SC; Cyan color: SW; Yellow color: QT); Boxes indicate the interquartile spread (25th and 75th quantiles) with the horizontal line indicating the ensemble median and the whiskers showing the extreme range of HWs frequencies and amounts in 29 cities Notes: There is no high temperature weather in which daily Tmax exceeds 35°Cin Kunming and Lasa cities in the past 60 years. Therefor there are 29 cities shown in this figure.

For advice 4, the title "2.2 Method" has been corrected into "2.3 Method"; the title "2.2.1 Heat wave index" has been corrected into"2.3.1 Heat wave index"; the title "2.2.2 Hot year index" has been corrected into"2.3.2 Hot year index".

Please also note the supplement to this comment: https://www.nat-hazards-earth-syst-sci-discuss.net/nhess-2019-335/nhess-2019-335-AC1-supplement.pdf

─────────────────────────

---

## Author Comment (AC2) · 26 Apr 2020

Dear reviewer,

Thanks very much for the review. The suggestions are very important for our manuscript. We have tried our best to revise the manuscript according your advices and explain the questions as much as we can. The details are listed as following which are also listed in the attached .pdf supplement file.

Best wishes,

Authors

(1) comments from Referees

The manuscript proposes new indices for the study of heat waves and extreme temperature especially for cities in China. I find the manuscript to be interesting and the methodology is novel. Unfortunately, the proposed methodology is not well descripted, and more details should be added. I find the overall information presented in this paper below the standards of the Natural Hazards and Earth System Sciences and I believe that the paper requires entire modifications and needs to go through the review process again.

Indeed, there are some aspects that are weak. The main problem is the proposed methodology for the new indices. More specific: âËŸA ′c The physical explanation of the index HWI (line 151, page 7) should be added. âËŸA ′c I believe that there is a mistake in the equation of the first index, HWI, (line 151, page7). Is the multiplication sign correct of the first parameter for the CD35? I can not understand why it is multiplication and not sum. The rate CD35/92 should be change to AD35/92. âËŸA ′c Moreover, I believe that an example should be added. I was tried to create an example for better understanding. Lets say that in a year there are 35 days with temperature greater than 35oC, from these days, there are 15 consecutive days with temp>35oC. Moreover from the initial 35 days, 15 days have temperatures greater than 37 oC (with 10 consecutive days greater than 37 oC) and 5 days have temperatures greater than 40 oC (with 10 consecutive days greater than 40 oC). Based on these data: HWI=(35/92 x 15/3+1)x(15/92+10/3+1)x(5/92+3/3+1) = HWI=(0.38x5+1)x(0.163+3.333+1) x(0.054+1+1)=2.9 x4.496x2.054=26.78 In case there is a mistake in the equation, HWI=6.38x4.496x2.054=58.92.

Based on the classification of Table 1, it is obvious that there is a mismatch for the range of the index. Please provide the appropriate modifications and explanations.

In the case of AHWI, there is a misunderstanding. It is not clear, how it is possible to be several HWI in a year. HWI use for its calculation the days with temperature greater

than 35 (37/40) in the three months (June, July, August 92 days). Based on it, it is not possible to have more than one value per year. Please give some explanations. Based on the above comment, HYI (line 173, page 8) can be not defined with the proposed way. Below the Authors can find some minor comments and suggestions in case of resubmission. âËŸA ′c Initially, I will suggest the description of the classification of the indices (table 1 and table 2) to be removed into methodology.

The analysis of figure 8 is not consistent with the figure 8. The scale of the diagram in figure 8 range from 0 to 900, the station Chongqing presents HWI equal to 800 while in the manuscript it is said ": : :sum value of HWIs in Chongqing reached 13.7: : :" (line 261, page 11). Similarly, the result about Changsha. Please made the appropriate modifications.

The section 3.1 can be changed to "variance of extreme temperature days" since in this paragraph it is analysed the trend of the extreme temperature days but the variance.

The quality of all figures is poor. The labels are too small, and it can not be read. âËŸA ′c The authors should add more information about the secondary axis in figure 4 and 7.

The authors claim that the analysis is for 31 main stations in China, in figure 3, 6 and 5 are presented the results of 29 stations, while in figure 7 and 8 are presented the result of 26 stations. Similarly, in map of figure 9 is presented 29 station. Please provide the appropriate modifications.

(2) Author's response:

1. The explanation of the index HWI had been described in the manuscript (line 143-148, page 6) . HWI in this manuscript is established mainly based on statistical and empirical methods, which is created to compare the intensities and frequencies of heat wave events. The physical mechanism of heat waves is not the focal point in this manuscript.

[Figure]

2. The example of HWI is added in the part of Heat wave index (line 156-166, page 7). The equation of HWI has been checked for several times. The multiplication sign of the first parameter for the CD35 is correct. There is no mistake in the HWI equation. The differences in the three parentheses of HWI equation are to distinguish the importance of CD35, AD37, CD37, AD40 and CD40. For heat wave events, 3 continuous days are the shortest duration for HWs, in which daily Tmax exceeds 35°C.In other word, the continuous days in which daily Tmax exceeds 35°Care the basic requirement of heat wave events. The HWI values are mainly determined by AD37, CD37, AD40 and CD40. The CD35 represents the continuous days in which daily Tmax exceeds 35°C, which is the same meaning of AD35. The discontinuous days in which daily Tmax exceeds 35°C are not belonged to heat wave events, which are not included in HWI equation; but these discontinuous days are exactly considered in HYI equation. According to the example that the reviewer had proposed, the value of CD35 should be 15, not 35; the values of AD37 and CD37 are 15 and 10; the values of AD40 and CD40 are 5 and 3. The calculation of HWI is, (15/92 x 15/3+1)x(15/92+10/3+1) x (5/92+3/3+1) =16.8.

3. The appropriate modifications on the classification of Table 1 has been done (line 553-554, page 21). We revise the description of each level of HWI, which becomes clear and easily understanding.

4. According to the definition of heat wave, the constant hot weather more than 3 continuous days in which daily Tmax exceeds 35°C could be called one heat wave event. If the days (daily Tmax≥35°C) are not continuous, it could not be named one heat wave event. There may be several heat wave events occurring in one year. For example, in 2014, there were two HWs occurred in Chongqing city, separately lasting from 17 July to 31 July and from 2 August to 8 August. So the HYI index should contain all the HWs and the discontinuous days with extreme temperature (daily Tmax exceeding 35°C) in one year. We have checked AHWI and HYI equations and there are no mistakes in them. We believe that the description of the classification of the indices (table 1 and table 2) should be in the part of Heat wave index (line 245-260,

page 11; line 289-292, page 12-13) , which are close to the analysis process of HW index and HYI index.

5. We have checked the analysis of figure 8; it is consistent with the figure 8. The scale of the diagram in figure 8 range from 0 to 900, which represents the sum value of HWIs of 60 years from 1955 to 2014. In order to make it clearer, the description in the manuscript has been changed (line 278-287, page 12).

6. We have checked the content of section 3.1. It contains variance of extreme temperature days and trend of the extreme temperature days. It is more proper to use the current title "3.1 Trend of Extreme Temperature days".

7. We have developed the quality of all figure to resolve the problems. More information about the secondary axis in figure 4 and 7 are added.

8. There are no high temperature weather in which daily Tmax exceeds 35°Cin Kunming and Lasa cities. So the results of the other 29 stations are presented in figure 3, 5 and 6. There are no HWs in Changchun, Shenyang, Guiyang, Kunming and Lasa cities. So the results of 26 stations are presented in figure 7 and 8. In order to be more clearer, we make appropriate modifications and add explanation in the titles of figure 3, 5, 6, 7 and 8.

(3) Author's changes in manuscript

For advice 1, there is no change in the manuscript. The interpretation has been given.

For advice 2, the example of HWI is added in the part of Heat wave index (line 156-166, page 7). The content is as follow:

For HWI, there are two extreme situations. If there are no heat waves in one year, the value of HWI would be 1. If there are 92 continuous days of a year in which Tmax exceeds 40°C, the value of HWI would reach the biggest, 33792; for the real world, the second extreme situation would rarely occur except extreme catastrophe shocking. According to the statistics from 1955 to 2014 in China, the most serious heat wave

event occurred in Changsha city in 2013 for which the value of HWI is no more than 140. The value of HWI is mostly determined by the number of continuous days in which Tmax exceeds 37°C, even 40°C. If the extreme hot days continue longer, HWI would be more serious. Taking the most serious heat wave event in Chongqing city for example, it lasted from 25 July to 19 August, 2006; the value of CD35 reaches 26; the value of AD37 is 21; the value of CD37 is 19; the value of AD40 is 9; the value of CD40 is 7. According to the HWI equation above, the HWI of this heat wave event reaches 98.2.

For advice 3, the description of each level of HWI (line 553-554, page 21) has been revised to make it clearer and easily understanding. The content is listed in the supplement file.

For advice 4, there is no change in the manuscript. The interpretation has been given.

For advice 5, in order to make it clearer, the description in the manuscript has been changed (line 278-287, page 12). The content is as following:

The sum value of HWIs in Chongqing is far bigger than other cities; the annual average value of HWIs in Chongqing reached 13.7. Changsha had been the second hard hit city with most serious HW hazards, in which the annual average value of HWIs reached 9.5. There were 6 cities that have been threatened by severer HW hazards, include: Hangzhou, Fuzhou, Nanchang, Xi'an, Wuhan and Haikou; the annual average value of HWIs in each city is between 4 and 9. There were 7 cities threatened by moderate severe HW hazards; these cities include: Hefei, Zhengzhou, Nanjing, Jinan, Shijiazhuang, Nanning, and Shanghai and the annual average value of HWIs in each city is between 2 and 4. The remaining 11 cities encountered lighter serious HW hazards in which the annual average value of HWIs is between 0 and 2. As mentioned above, there were no HW hazards in 5 cities.

For advice 6, there is no change in the manuscript. The interpretation has been given.

[Figure]

For advice 7, we have added more information of the secondary axis in figure 4 and 7 and developed the quality of figures. The figure 4 and 7 are shown in the supplement file.

For advice 8, appropriate modifications and explanation in the titles of figure 3, 5, 6, 7 and 8 have been added. The content is as following:

Fig.3 Distribution of D35 in 29 cities from 1955 to 2014 (Green color: NE; Blue color: NW; Red color: NC; Purple color: CC; Black color: EC; Orange color: SC; Cyan color: SW; Yellow color: QT); Boxes indicate the interquartile spread (25th and 75th quantiles) with the horizontal line indicating the ensemble median and the whiskers showing the extreme range of D35 in 29 cities

Notes: There are no high temperature weather in which daily Tmax exceeds 35°Cin Kunming and Lasa cities in the past 60 years. Therefor there are 29 cities shown in this figure.

Fig.5 Comparison between D35 and HWs per year in 29 cities of China from 1955 to 2014

Notes: There is no high temperature weather in which daily Tmax exceeds 35°Cin Kunming and Lasa cities in the past 60 years. Therefor there are 29 cities shown in this figure.

Fig.6 Distribution of amounts and frequencies of HWs in 29 cities from 1955 to 2014 (upper graph: amounts of HWs; lower graph: Frequency of HWs. Green color: NE; Blue color: NW; Red color: NC; Purple color: CC; Black color: EC; Orange color: SC; Cyan color: SW; Yellow color: QT); Boxes indicate the interquartile spread (25th and 75th quantiles) with the horizontal line indicating the ensemble median and the whiskers showing the extreme range of HWs frequencies and amounts in 29 cities

Notes: There is no high temperature weather in which daily Tmax exceeds 35°Cin Kunming and Lasa cities in the past 60 years. Therefor there are 29 cities shown in

this figure.

Fig.7 Frequency of Low, Moderate, High and Extreme high HW hazards in 26 cities from 1955 to 2014 (Top left: NC; Top right: SW & SC. Middle left: NE; Middle right: NW & QT; Bottom left: CC; Bottom right: EC)

Notes: There are no HWs in Changchun, Shenyang, Guiyang, Kunming and Lasa cities in the past 60 years. Therefor there are 26 cities shown in this figure.

Fig. 8 The sum values of HWIs in 26 cities from 1955 to 2014

Notes: There are no HWs in Changchun, Shenyang, Guiyang, Kunming and Lasa cities in the past 60 years. Therefor there are 26 cities shown in this figure.

Please also note the supplement to this comment:
https://www.nat-hazards-earth-syst-sci-discuss.net/nhess-2019-335/nhess-2019-335-AC2-supplement.pdf

---

## Author Response (AR2)

**Comments from referee:**

This paper is the revised manuscript of the study entitled "Spatiotemporal Changes of Heat Waves and Extreme Temperatures in Main Cities of China from 1955 to 2014" (manuscript ID: nhess-2019-335). The authors followed many of the reviewer's commends and the structure of the manuscript are improved. I have only some minor comments Finally, I believe that the revised manuscript accomplishes the Journal's tasks and it can be published. More specific:

• In the physical explanation of the index HWI (line 156-166, page 7) following the presented example the HWI=(26/92 x 26/3+1)x(21/92+19/3+1)x(9/92+7/3+1) = 89.5. There is a mismatch between the two values (98.2 and 89.5), please provide the appropriate modification.

• The selection of the thresholds used in classification of the HWI, is subjective. It could have an impact on the result's robustness. The annual average value of HWIs in Chongqing and Changsha is 13.7 and 9.5 respectively, which are much higher than the highest threshold of Table 1 (HWI>6). Moreover, using the previous example (line 156-166, page 7) the magnitude of the HWI is extremely higher than the highest threshold of Table 1 (HWI>6). For that reason, I strongly suggest to use an objective classification. The authors could make a distribution analysis of the HWI and then they can estimate the 1st (25% percentile), 2nd (50% percentile) and 3rd quantile ((75% percentile) and use them as thresholds.

HWI =1.0 No hazard

1.0 < HWI≤1st quantile Low hazard

1st quantile < HWI≤2nd quantile Moderate hazard

2nd quantile < HWI≤3rd quantile High hazard

HWI>3rd quantile Extreme high hazard

Figure 7 and its analysis should change accordingly.

• Finally, the quality of all figures remains poor. In the majority of the figures, the labels are too small, and it can not be read.

**Author's response:**

1. The value of HWI (line 163-166, page 7) has been modified into 89.5.

2. Based on the suggestion of referee, the thresholds used in classification of the HWI (Table 1, line 554-555, page 21) have been changed according to percentiles, such as 1st (50% percentile), 2nd (75% percentile) and 3rd quantile (95% percentile). The classification is as follows:

HWI =1.0                                    No hazard

1.0 < HWI≤1st quantile                  Low hazard

1st quantile < HWI≤2nd quantile       Moderate hazard

2nd quantile < HWI≤3rd quantile       High hazard

HWI>3rd quantile                         Extreme high hazard

Accordingly, Figure 7 (line 595-601, page 25) and its analysis (line 252-275, page 11-12) have been changed.

3. The quality of all figures has been checked and the figures with low quality have been improved.

**Author's changes in manuscript**

For advice 1, the changes in line 163-166, page 7 are as follows:

Taking the most serious heat wave event in Chongqing city for example, it lasted from 25 July to 19 August, 2006; the value of $CD_{35}$ reaches 26; the value of $AD_{37}$ is 21; the value of $CD_{37}$ is 19; the value of $AD_{40}$ is 9; the value of $CD_{40}$ is 7. According to the HWI equation above, the HWI of this heat wave event reaches 89.5.

For advice 2, the changes in Table 1 (line 554-555, page 21), Figure 7 (line 595-601, page 25) and its analysis (line 252-275, page 11-12) are as follows:

Tab.1 The classification of HW hazards by the values of HWI

| Heat Wave Index | Level of hazard | Description |
|---|---|---|
| HWI =1.0 | No hazard | There is no HW event occurred. |

| | | |
|---|---|---|
| 1.0 < HWI ≤ 1.13 | Low hazard | The HW event must last at least 3 continuous days and less than 6 continuous days, in which there is no days above 37℃ or 40℃. |
| 1.13 < HWI ≤ 1.99 | Moderate hazard | The HW event must last at least 3 continuous days and less than 17 continuous days, in which daily Tmax exceeds 35℃. |
| 1.99 < HWI ≤ 4.83 | High hazard | The HW event must last at least 3 continuous days and less than 21 continuous days, in which daily Tmax exceeds 35℃. |
| 4.83 < HWI | Extreme high hazard | The HW event must last at least 3 continuous days in which daily Tmax exceeds 40℃. |

[Figure]

[Figure]

[Figure]

Fig. 7 Frequency of Low, Moderate, High and Extreme high HW hazards in 26 cities from 1955 to 2014 (Top left: NC; Top right: SW & SC. Middle left: NE; Middle right: NW & QT; Bottom left: CC; Bottom right: EC)

*Notes: There are no HWs in Changchun, Shenyang, Guiyang, Kunming and Lasa cities in the past 60 years. Therefor there are 26 cities shown in this figure.*

The thresholds of 5 HWI levels are separately determined by 50% percentile, 75% percentile, 95% percentile of all heat wave events which occurred in the past 60 years. When the value of HWI is 1.0, it indicates that there is no continuous hot day in which Tmax exceeds 35℃. When the value of HWI is between 1.0 and 1.13, it indicates slight HW hazards in which the duration and intensity of HWs are minimal. When the value of HWI is between 1.13 and 1.99, it means HW hazards are slight as there are few continuous days of Tmax exceeding 37℃. When the value of HWI is between 1.99 and 4.83, it indicates that the HW hazards are serious and the continuous days of

Tmax exceeding 37℃ or 40℃ become frequent. When the value of HWI is above 4.83, it indicates
that the HW hazards are very serious and the continuous days of Tmax exceeding 37℃ or 40℃
may last through the whole period of HWs.
According to the classification of HWI, the frequencies of HW hazards with different levels in the
past 60 years in 31 typical cities of China are analyzed (Fig.7). In all, cities with low HW hazards
were the majority accounting for 52.9% of all HWs; the moderate HW hazards accounted for 22.3%;
the high HW hazards represented 19.8%; and the extreme high HW hazards accounted for 5.0%.
For all the 31 cities, most of the HW hazards are not serious; only 1/20 of the HW hazards are of
the greatest threats. No HW hazards occurred in Changchun, Shenyang, Guiyang, Kunming and
Lasa from 1955 to 2014; no high or extreme high HW hazards occurred in Haerbin, Xining,
Yinchuan and Chengdu; no extreme high HW hazards occurred in Beijing, Tianjin, Taiyuan,
Huhehaote, Wulumuqi, Lanzhou and Guangzhou; in the remaining 15 cities, there were all four
levels of HW hazards occurred in the past 60 years. However, most HW events of high (1.2 per
year) and extreme high (0.6 per year) levels occurred in Chongqing than the other cities; most HW
events of moderate levels occurred in Changsha, reaching 1.0 per year; and most HW events of
low level occurred in Haikou, reaching 2.3 per year.
For advice 3, all figures have been checked. Figure 7 has been improved according to the
referee's suggestion. All the other figures remain a stable quality. When the figures are enlarged,
the labels and numbers in the figures are clear enough, especially in the .doc or .docx versions.

[revised manuscript text omitted]